# Inversion Table Fall Injury, the Phantom Menace: Three Case Reports on Cervical Spinal Cord Injury

**DOI:** 10.3390/healthcare9050492

**Published:** 2021-04-22

**Authors:** Seung-Hwan Jung, Jong-Moon Hwang, Chul-Hyun Kim

**Affiliations:** 1Department of Rehabilitation Medicine, Kyungpook National University Hospital, Daegu 41944, Korea; pyromyth@naver.com; 2Department of Rehabilitation Medicine, School of Medicine, Kyungpook National University, Daegu 41944, Korea

**Keywords:** cervical spine injury, tetraplegia, traction, exercise equipment, safety measure

## Abstract

Background: An inversion device, which is used to suspend one’s body and perform traction therapy, was introduced as an inversion table under the name of “Geokkuri” in South Korea. Fall injuries while hanging on inversion tables are among the most devastating spine injuries, as the likelihood of severe neurological sequelae such as tetraplegia increases. However, its enormous danger has been overlooked and this devastating injury has become a common clinical entity over time. The limited number of studies reported imply the lack of interest of researchers in these injuries. We reviewed three cases of spinal cord injury sustained on inversion tables in different environments and report the potential danger associated with the use of inversion tables to facilitate a safer exercise environment.

## 1. Introduction

Traction is a type of physical therapy that involves suspending the body at an angle, or completely upside down, to allow the vertebral bodies to separate, facet joints to glide, intervertebral foramen to widen, and spinal curves and musculature to be stretched [1,2,3]. Traction is well known for treatment of lumbar discogenic disease, which allows negative pressure to pull the herniated fragment back into the disc space [4,5]. The inversion device was introduced as “Geokkuri” in South Korea; this device allows performing traction therapy with ease at home. Public exercise equipment or fitness trails in South Korea are commonly equipped with inversion devices (Figure 1). You lie on the inversion table, lock your ankles, and gently begin to lean back onto the table and then invert yourself further (Figure 2).

It is unclear if inversion tables actually relieve lower back pain. Some people may find traction temporarily helpful, but there is a debate about the effectiveness of traction [6,7,8]. Among the studies on the effectiveness of traction, traction type or dosage is heterogeneous, which could influence the effectiveness. Additionally, there are several types of tractions such as continuous or intermittent, and also tractions can be mechanical, motorized, or manual. Beurskens AJ et al. described traction effect as placebo, when traction forces loaded less than one-fifth of the body weight [9], whereas others insisted that this can also be effective [6,7].

Health risks may be relatively higher compared to the questionable benefits. Even falls from short distance can also cause serious injuries, most commonly to the head and cervical spine [10,11]. It is clear that many of these devices are not safe under certain circumstances. Certain designs, specifically the single-pin ankle lock design that is commonly found in public devices (Figure 3), does not have double lock or additional safety device. While suspended upside down vertically, the user should bear one’s weight against single-pin lock entirely. If the feet slip off the user will fall and possibly suffer a severe cervical spine injury, which can be life threatening. Anyone can make mistakes and can fall, but in this case the consequences are irreversible. Falls from 1 or 2 feet high are by no means negligible.

In the following case series, we present cases of both complete and incomplete cervical spinal cord injuries from inversion table falls. From 2015 to 2019, three patients were admitted to our rehabilitation unit. Accidents occurred in different places, such as at home, on outdoor exercise equipment, or on a hiking trail, but the mechanism of the injuries were identical—their feet or ankles slipped, and they fell to the ground. We report the potential dangers associated with the use of inversion tables, facilitate safer utilization, and support policy development in the future. We would like to emphasize that, the point is, all three cases were related to the correct use of the device, not improper use. The case report was approved by the Institutional Review Board of Kyungpook National University Hospital. The patients signed an informed consent for the publication of this report.

## 2. Case Report 1

A 50-year-old healthy woman without any relevant medical history went to a nearby park with public outdoor exercise equipment on the morning of 19 October 2019. The patient had been previously active and well before the accident. The patient was working out with the public exercise equipment. While hanging on the inversion table upside down, she slipped and hit her head on the ground. Thereafter, she collapsed and immediately experienced a total loss of motor power in her whole body; she was found at the park and directly transferred to our hospital by ambulance. Computed tomography (CT) brain scan results did not reveal any abnormalities. Cervical spine CT scan showed a bilateral facet dislocation on C5-6 and anterior displacement of the C5 vertebral body (Figure 4a). Cervical spine magnetic resonance imaging (MRI) also showed an anterior displacement of C5 on C6, a complete discoligamentous complex (DLC) injury, which was causing cord compression with extensive edematous cord signal change, and an intramedullary hemorrhage (Figure 4b).

During the examination, she was conscious with a Glasgow coma scale score of 15 out of 15. The patient’s body weight and height were recorded as 76 kg and 163 cm, respectively, with a BMI of 28.6. Cardiovascular and respiratory examinations showed hypotension with desaturation, which was considered to be caused by spinal shock and high-level cord injury. The emergency team applied inotropic agents and mechanical ventilator care. Her lower limbs were flaccid with a power grade of 0, while upper limbs showed a power grade of 3 on the shoulder, elbow, and wrist; however, her fingers were also flaccid with a power grade of 0. Knee and ankle reflexes were diminished with all sensory modalities below the neurological level of injury decreased. Anal tone was flaccid, with sensory function preserved yet hyposensitized. All examination findings diagnosed her as incomplete spinal cord injury of American Spinal Injury Association Impairment Scale (AIS) grade ‘B’, according to International standards for neurological classification of spinal cord injury (ISNCSCI).

Urgent neurosurgical referral was provided, and the patient had undergone anterior cervical discectomy and fusion (ACDF) of C5-6 and posterolateral mass screw fixation of C5-6. After post-operation care, she was referred to the Department of Rehabilitation Medicine on 30 October 2019.

The patient received multidisciplinary rehabilitation therapy with conservative treatments for her symptoms. Interventions included standing on tilt table treatment, passive range of motion exercise therapy, sitting balance improving training, arm strengthening exercise, and functional electrical stimulation. Therapy session consisted of 30 min of complicated rehabilitation therapies for two sessions a day. As the quality of life deteriorated after the injury, she experienced depressive mood changes and was prescribed antidepressant after consulting a psychiatrist. The last follow-up assessment before discharge was performed on 12 December 2019, and the physical examination showed little improvement. Her manual muscle test results remained unchanged. Her Berg Balance Scale (BBS) score was still zero. The patient had failed to void and an urodynamic study diagnosed her as an acontractile type neurogenic bladder. After 44 days of inpatient intensive rehabilitation therapy, she was transferred to another rehabilitation hospital for long-term rehabilitative management.

## 3. Case Report 2

A 63-year-old healthy woman went to a nearby public health center with public outdoor exercise equipment on 23 March 2018. She had undergone right total mastectomy due to breast cancer in 2000 and had been active and capable of performing daily living activities before the accident. At the public health center, she tried the inversion table. Hanging on the inversion table upside down, her feet slipped out of her shoes, and she fell and hit her head on the ground. The patient collapsed and immediately experienced a total loss of motor power in whole body; she was found and transferred to the emergency room of a local medical center. Non-enhanced brain CT did not reveal any abnormalities. Cervical spine CT revealed dislocation of C5-C6 (Figure 5a). For emergent surgical management, the patient was transferred to our hospital. Cervical spine MRI revealed flexion distraction with translation injury at C5-6, leading to C5-6-7 cord contusion with an intramedullary hemorrhage and complete DLC disruption. (Figure 5b).

During the examination, she was conscious and had a Glasgow coma scale score of 15 out of 15. The patient’s body weight and height were recorded as 65 kg and 158 cm, respectively, with a BMI of 26.04. Cardiovascular and respiratory examinations showed orthostatic hypotension. During 3-position blood pressure measurement, her systolic pressure decreased from 133 to 74 mmHg in 3 min when she changed her position from supine to 45° tilt. Power of the upper limbs decreased to grade 3, while the lower limbs were flaccid with a power grade of 0. Tones and reflexes of the biceps, knee, and ankle were diminished due to spinal shock. All sensory modalities below the C7 sensory level decreased. Anal tone was flaccid with preserved sensory function, and all examination findings indicated a diagnosis of incomplete spinal cord injury of AIS grade ‘C’ according to International Standards for Neurological Classification of Spinal Cord Injury (ISNCSCI).

Urgent neurosurgical referral was provided, and the patient underwent posterior open reduction and ACDF of C5-6-7. After post-operation care, she was referred to the Department of Rehabilitation Medicine on 2 April 2018.

The patient received multidisciplinary rehabilitation therapy with conventional conservative care for her symptoms. Interventions included standing on tilt table treatment, assisted exercise therapy, sitting balance improving training, arm and trunk strengthening exercise, and also functional electrical stimulation. Therapy session consisted of 30 min of complicated rehabilitation therapies for two sessions a day. She applied an abdominal binder and took anti-hypotensive medications to treat orthostatic hypotension. She had neuropathic pain in her hands and was prescribed 300 mg of gabapentin three times a day. Follow-up physical examinations showed minimal improvement. The last follow-up assessment before discharge was performed on 21 April 2018, and her neuropathic pain relieved as the numeric pain rating scale (NRS) score decreased from 3 to 1; her motor power grade in the proximal part of the upper limb improved from 3 to 3+. Motor power in the lower limb remained similar compared to initial physical examination and her Berg Balance Scale (BBS) score remained zero. After 20 days of inpatient intensive rehabilitation therapy, she was transferred to another local rehabilitation center for long-term rehabilitation therapy.

## 4. Case Report 3

A 58-year-old healthy woman without any other medical history was doing at-home workout in her own house on 31 January 2015. The patient had been previously active and well before the accident. After training, she was hanging upside down on her inversion table. While hanging on the inversion table, she slipped and fell, hitting her head on the ground. She collapsed and immediately experienced a complete loss of body motor function. She visited the emergency room of a local medical center. Cervical spine CT revealed a bursting fracture on C6 (Figure 6a). For further evaluation, she was transferred to our hospital. Cervical spine MRI revealed heterogeneous signal change and intramedullary hemorrhage introducing compressive cervical myelopathy. (Figure 6b).

During the examination, she was conscious with a Glasgow coma scale score 15 out of 15. She was 157 cm tall, weighed 67 kg, and had a BMI of 27.18. Her both legs were flaccid with a motor grade of 0, while the upper limbs showed power grade of 3- on the shoulder, elbow, and wrist. However, her fingers were flaccid with a power grade of 0. Tones and reflexes of the biceps, knee, and ankle were diminished due to spinal shock. All sensory modalities below the C6 sensory level were absent. More importantly, with absence of anal sphincter contraction and perianal sensation, she was diagnosed as complete spinal cord injury of AIS grade ‘A’ according to ISNCSCI.

Urgent neurosurgical referral was provided, and anterior corpectomy on C6 with ACDF of C5-6-7 and posterolateral mass screw fixation of C5-6-7 were performed. She presented severe hypotension with fever which was caused by spinal shock and septic shock. Conservative post-operation care with intravenous inotropic agents and antibiotics was administered, and she was referred to the Department of Rehabilitation Medicine on 26 February 2015.

Like two cases, she received multidisciplinary rehabilitation therapy for two sessions a day in our rehabilitation unit. Interventions also included standing on tilt table treatment, passive exercises on both legs, arm strengthening exercises, and functional electrical stimulation. She presented neuropathic pain in whole extremities and took 300 mg of gabapentin three times a day. Since she showed depressive mood changes after the injury, we prescribed antidepressant after consulting a psychiatrist. Follow-up assessments did not show any improvement. The last follow-up assessment before discharge was performed on 29 March 2015 and her neuropathic pain had improved as NRS score decreased from 5 to 3, while her motor power grade remained unchanged. Her BBS score improved from 0 to 2. An urodynamic study conducted before discharge showed neurogenic bladder with low detrusor pressure and detrusor external sphincter dyssynergia. She failed to void and an indwelling catheter was inserted, and she was transferred to local rehabilitation center for long term treatment.

## 5. Discussion

The inversion table is widely used in South Korea for exercise and to relieve lower back pain. Most public outdoor exercise places are equipped with inversion tables. Inversion tables can also be seen in parks, hiking trails, public health centers, or gyms. Similar to that seen in one of the cases, some people have inversion tables at their homes. This means they are easy to access and familiar to the public. However, the danger associated with inversion tables is rarely known.

When a person hangs upside down on the inversion table, the center of the body is about three to four feet high from the ground. Even in the inverted position, one’s vertex is a few inches above the ground. It does not seem to be dangerous, but it is not true. The mechanism of inversion table fall injury resembles that of diving injury—once the ankle power is lost or it slips, the user will land on top of their head. Like head first entry into shallow water, even a fall from a couple of feet high can cause devastating cervical spine injury [12,13]. The potential energy of one’s weight is transformed into kinetic energy and then transmitted to the cervical spine in the form of vertical compression. Nightingale, R.W. et al. described more than half of the traumatic cervical spine injury sustained from a form of bilateral facet dislocation, which most commonly occur at C5 and C7 level. The primary causes of catastrophic cervical spine injuries are compressive buckling and compression. However, it does not imply that natural curvature of the neck was eliminated prior to the injury in the form of bending, neither flexion nor extension first [14]. 3.2 m/s is a threshold velocity which is known to cause these injuries to be the threshold velocity under compression and buckling scenario [12]. However, several impact experiments suggested that the actual threshold that produces catastrophic neck injuries could be lower than 3.2 m/s [15,16,17,18]. Bauze and Ardran revealed that, as we can see in these three cases, when the crown of the head is locked on the ground the dislocation of the cervical spine requires much less force, which can occur without fracture [19]. An epidemiological study conducted by Silver explained that the cervical spine is particularly vulnerable due to its mobility and the disparity in the movement of unsupported skull on the cervical vertebrae. A flexion force exerted on the cervical spine through the head crushes the vertebral body and extrudes the disc and vertebrae into the spinal cord [20].

All three patients underwent anterior approach, cervical discectomy and fusion (ACDF) surgery as a surgical treatment in the acute phase. Although there are still debates about the management of cervical dislocation and associated injuries, recent studies revealed anterior approach and fusion with corpectomy if needed is an effective surgical technique for the treatment of cervical spondylosis accompanying spinal cord injury [21,22].

Clinical characteristics of three patients were summarized in Table 1. In this case series, all three patients were middle-aged and obese. It is hard to generalize; however, middle-aged or elderly women have weaker muscle power, particularly in the ankle dorsiflexion compared to that of the general population [23,24,25]. All the patients were obese, with BMI over 26, which means that there were strong compressive forces in play when the patient collapsed. In order to investigate whether obesity, gender, or age have a significant correlation with the fall while using inversion table, future research regarding a huge sampling of similar accidents or model experiments is necessary. The type of the involved surface was omitted in our study. In future research, it is recommended to include the type of the surface (lawn, concrete floor, dirt floor, etc.) since it is an important factor when it comes to the collision and energy transfer.

## 6. Conclusions

Due to the danger of inversion table devices, we believe that some modifications should be introduced, but the changes are yet to be made. There are several alternatives to make these devices safe. First, a strap can be attached to hold the ankles and feet tighter and to prevent users from slipping down. In addition, an angled pillow or rubber ramp can be positioned just below the user’s head, allowing the user to slide to the floor rather than fall directly on their head.

Milner-Brown, H. S. et al. described quantification of human muscle strength, endurance and fatigue. According to this study, the ankle dorsiflexors showed 34% of maximum force reduction after 1 min duration isometric force [26]. To prevent fall accident from the inversion table, it is recommended to reduce usage time less than 1 min. The users are encouraged to take at least 1.5 min recovery time between the trials.

Since the nature of inversion table “Geokkuri” is familiar to the general public and easily accessible, the dangerous situations have been underestimated. The users, especially aged or obese female users, who utilize inversion tables should be cautious to avoid falling.

## Figures and Tables

**Figure 1 healthcare-09-00492-f001:**
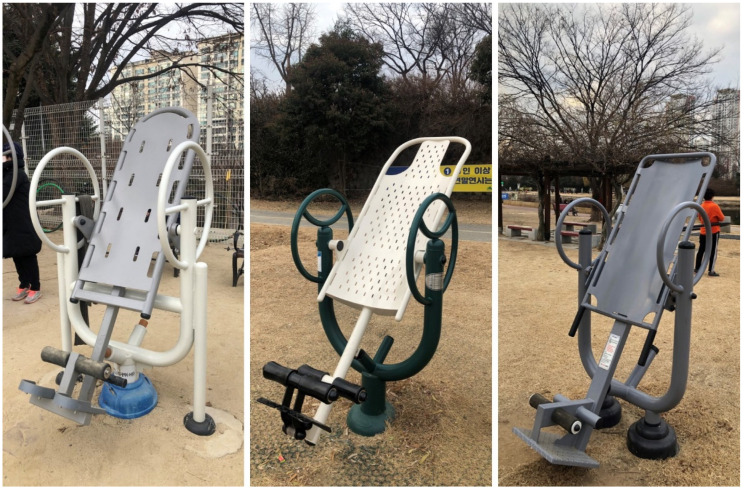
Inversion devices equipped in public exercise places in South Korea.

**Figure 2 healthcare-09-00492-f002:**
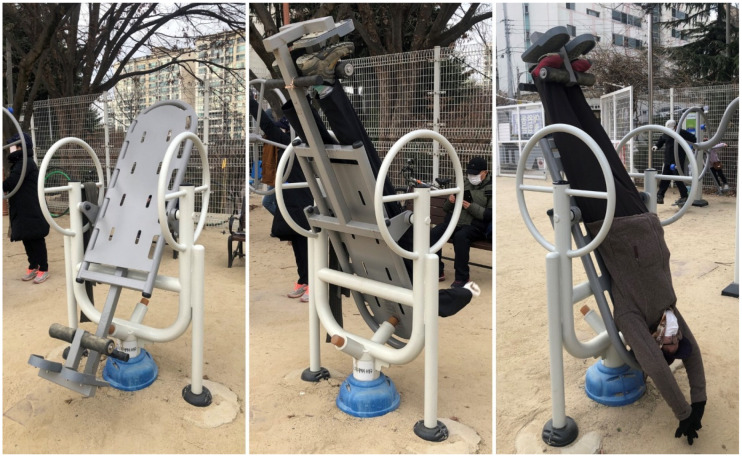
The user lies on the inversion table and invert oneself vertically.

**Figure 3 healthcare-09-00492-f003:**
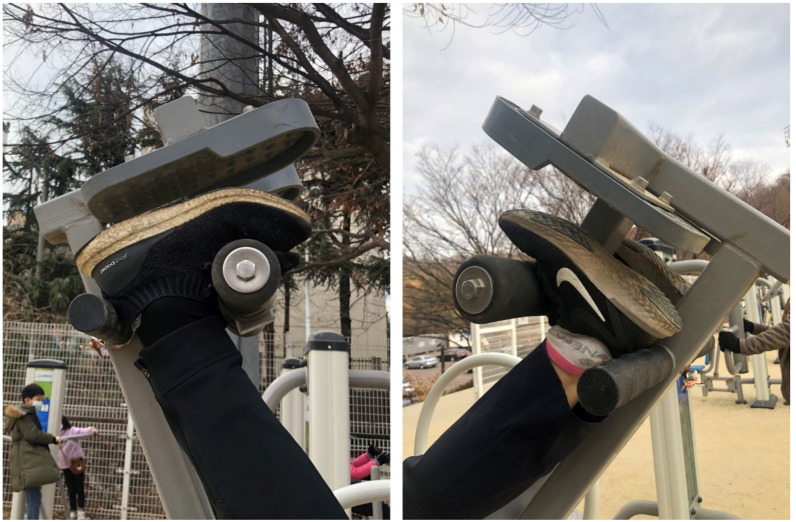
Single-pin ankle lock design is commonly found in public devices.

**Figure 4 healthcare-09-00492-f004:**
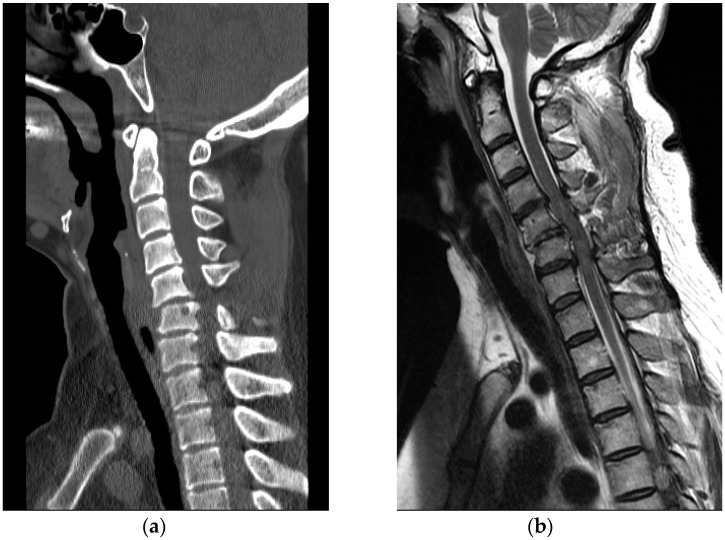
(**a**) Cervical spine CT revealed facet dislocation at C5-C6 and anterior displacement of the C5 vertebral body; (**b**) Cervical spine MRI showed cord compression with edematous cord signal change and an intramedullary hemorrhage.

**Figure 5 healthcare-09-00492-f005:**
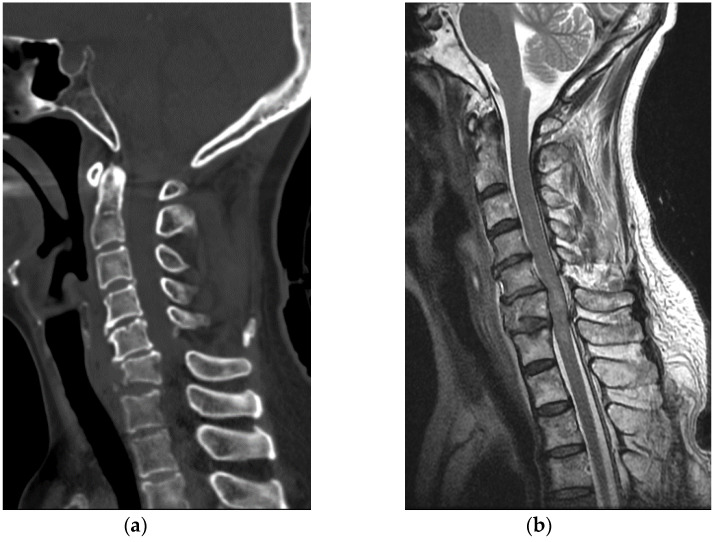
(**a**) Cervical spine CT showed C5-C6 dislocation with C6 vertebral body compression fracture; (**b**) Cervical spine MRI revealed flexion-distraction with translation injury at C5-C6, cord contusion with an intramedullary hemorrhage and complete DLC disruption.

**Figure 6 healthcare-09-00492-f006:**
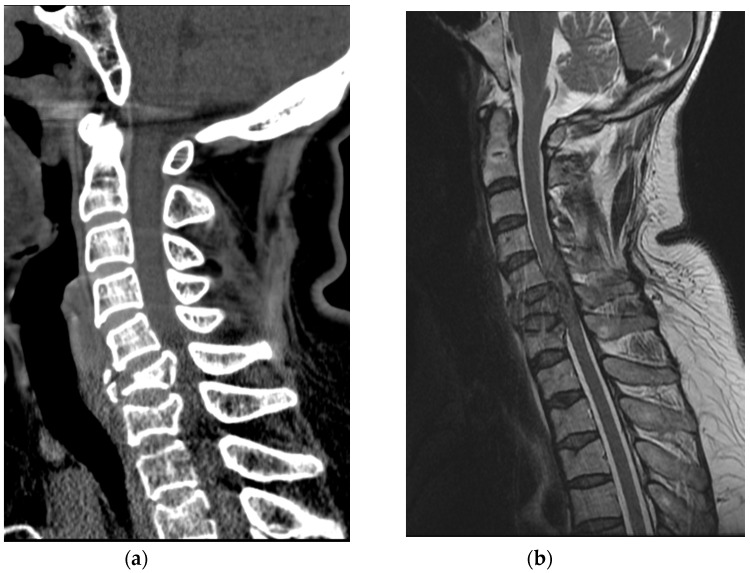
(**a**) Cervical spine CT showed C6, C7 vertebral body bursting fracture; (**b**) Cervical spine MRI revealed cord compression and heterogeneous signal change with hemorrhagic infiltration.

**Table 1 healthcare-09-00492-t001:** Summary of the clinical characteristics in described cases.

	Case 1	Case 2	Case 3
Age/Gender	50/F	63/F	58/F
BMI	28.6	26.04	27.18
Location	Public exercise equipment	Public health center	Home
Diagnosis	Facet dislocation at C5-6	C6 compression fracture & C5-6 dislocation	C6, C7 bursting fracture
Initial motor power			
U/E	3	3	2-
L/E	0	0	0
NLI	C5/C5	C6/C6	C6/C6
AIS classification	B	C	A

BMI: body mass index, U/E: upper extremity, L/E: lower extremity, NLI: neurological level of injury, AIS: American spinal injury association impairment scale.

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
