# Peer review of "Inversion Table Fall Injury, the Phantom Menace: Three Case Reports on Cervical Spinal Cord Injury"

_healthcare, 2021, doi:10.3390/healthcare9050492_

Round 1
Reviewer 1 Report
The authors solved my criticisms. Accept in present form
Author Response
Response to Reviewer 1 Comments
Dear Reviewer #1,
Thank you very much for your kind letter and comments concerning our manuscript entitled “Inversion Table Fall Injury, the Phantom Menace: Three Case Reports on Cervical Spinal Cord Injury”.
We are very pleased to hear from you that our response solved your criticisms. With your sincere advice, this study is able to convey the contents more clearly. Thank you again for your detailed opinions about the manuscript.
Reviewer 2 Report
The manuscript “Inversion Table Fall Injury, The Phantom Menace: Three Case Reports on Cervical Spinal Cord Injury”, by Jung et al., deals with the fact that a device supposed to serve the healthcare of people is actually sometimes causing very serious injuries.
The device is an inversion table introduced in South Korea. The manuscript is a Case Reports: three cases of spine injuries due to falls from the device are reported. The authors provide a complete description, for each case, of the condition of the patient after the accident, the adopted therapy, and the conditions after the therapeutical period.
In my opinion the manuscript is an interesting report about something that deserves public interest. It may happen that a tool or a device can be improperly used and can cause accidents: but in this case we are talking about very serious accident with permanent (still serious) consequences, and related to an essentially correct use of the device (standing with the body reversed is the reason to use the device!). As a first suggestion, my advice for the authors is to emphasize this point. They are not talking about accident by improper use, but accident related to the main idea of the device.
While reading, my attention was captured by the fact that the three reported cases concerns aged and over-weight people. The authors actually underline this circumstance at the end of the manuscript, as these can be (quite obviously) a risk condition. However, it would be nice to have a huger sampling, only if possible, on this detail. Are there reported, in the clinical cases in South Korea or elsewhere, other occurrences of similar accidents from the use of the same device? The authors would not need, of course, to describe clinical details of cases not concerning their healthcare structure, but just mention if the “distribution” of patient is actually biased in this sense. I do not know if such an information is accessible – if yes, please report. In addition: do the authors know if the company producing the device show any warning or danger ad for over-weight people?
At the beginning of the discussion section, I would suggest to put a short table summarizing the described cases (maybe just few columns with initial condition, therapy, final conditions, remarks and follow-ups).
I appreciated the suggestions that the authors gives in the conclusion section. They are indeed very simple suggestions: a pillow, or a strap to hold the ankles. The latter was also my thought, and I am extremely (and negatively) surprised to learn that such a safety accessory is not provided. According to the experience of the authors, would it compromise the effectiveness of the body stretching? Probably, an interaction with the company about the reason for not having included such safety strap is beyond the purpose of the paper; but, still in the knowledge of the authors, do similar devices include a safety belt or something like that? Do the authors know if there is any law concerning this issue (such a law, if any, would be probably fundamental for the effectiveness of the manuscript in preventing further similar accidents).
Reviewer 3 Report
The present article describes 3 case studies where females slipped down from an inversion table and sustained serious spine injuries after making contact with the ground. The introduction is well written and very nice visuals are provided that really make the study clear. The discussion could use some clarifications and details. But overall, it is a quality and interesting study.
Minor comments are below:
Place refs 3-4 in line 38 also.
Line 45 – this should be stated less certain (eg, health risks may be relatively …). If this is a stated fact, then please provide a couple good references. Alternatively, you could justify this more by describing how a fall from only 1 or 2 CAN cause permanent damage and even death. For example, in the Discussion, many references are listed. I would imagine a couple should be included here to support the possible cervical damages that could result from a drop from a small distance.
Line 52 – do not say ‘happening” but rather “falls” or “mishaps”. Also, 6 feet seems like a very large distance to fall. The pictures show a head drop of only about 1 or 2 feet. Please clarify.
Great images in Figure 4.
Line 139 – please spell out ISNCSCI
Line 155 – please spell out “BBS”
Good job describing the mechanisms of injury in Discussion. On Line 213, the authors note compressive buckling and compression as the main mechanism “similar to diving injury”. Can the authors confirm if the “compression” implies that the neck’s natural curvature was eliminated prior to injury (eg, neck flexion first). Also, can the authors add a sentence to describe why other mechanism are unlikely (eg, neck extension + hyper-extension).
Line 214 – This study referenced [5] is good to highlight more because the impact velocity of 3.2 m/s is about the same velocity that would be obtained by a projectile dropped from about 1.5 ft. This distance corresponds well to the present study. However, at least a few sentences about the surface where the head contacted would be good. In particular, the impulse and momentum equation is directly linked to the time of deceleration and thus the surface involved. For example, grass vs. concrete makes gigantic difference.
Line 222 – please clarify the sentence and word “discrepancy”.
Line 225 – do not start a sentence with “And about …”
Line 231 – please provide a reference for this statement
Where there any reports on how long the person maintained the reclined position? Perhaps there a limit in time that would be reasonable to suggest as the dorsiflexors fatigue?
Line 237 – Please fix the first couple sentences in Conclusions due to poor grammar.
Reviewer 4 Report
Title: Inversion Table Fall Injury, The Phantom Menace: Three Case Reports on Cervical Spinal Cord Injury
I carefully read the manuscript.
The major concern of the study is that the authors did not clearly present the reason or mechanism why Geokkuri was used for SCI. I worry that this study was not scientifically conducted.
1) Introduction
The introduction is very short and lack of explanation on why Geokkuri was used for SCI, it is suggested to make a more in-depth review, since there are only 4 citations to traction, which indicates that with only 4 studies the background and scientific evidence of the study topic has not been sufficiently collected.
2) Materials and Methods
- The sample case is very small, it is suggested to expand the RCT. This is considered a very important handicap and one that must be resolved. The case study is an experimental method that examines the change of the subject while the treatment is the cause as intervention occurs from the base-line to the treatment line. For scientific case studies, it is suggested to use the methods of ABA design and ABAB design.
- Geokkuri has a safety problem to apply to SCI patients. The ankle must bear the weight of the body weight, but the SCI patient cannot support the weight due to the lack of muscle strength in the lower extremities due to paralysis. The subject's lower extremity muscle strength is grade 0 in Case Report 1.
- Before applying Geokkuri to a patient, verification of the traction effect through an engineering mechanism must be preceded, but such a part is insufficient. It is not suitable to simply compare Geokkuri with the traction paper. Traction is performed in sitting & supine posture to ensure patient safety, but Geokkuri does not have a minimum safety device to apply to SCI.
3) Discussion
- Discussion is important in the article, but this paper rarely discusses the results. Depending on the research method, a discussion on each result should be developed.
4) Bibliographic references
- It must be expanded and updated. I have only found 5 quotes from the last 10 years. And, among the cited papers, there are no papers related to traction within the last 10 years.
Therefore, this manuscript is not of sufficient quality to be published in Healthcare.
Round 2
Reviewer 4 Report
I think that the manuscript revised by the author is appropriate. Therefore, I suggest that the revised manuscript should be accepted as in the current form.